# Response of blueberry photosynthetic physiology to light intensity during different stages of fruit development

**Jia Long[1], Tianyu Tan[2], Yunzheng Zhu[1], Xiaoli An[1], Xinyu Zhang[1], Delu Wang[1]***

**1** College of Forestry, Guizhou University, Huaxi, Guiyang, Guizhou, China, **2** Forestry Bureau of Kaili, Kaili, Guizhou, China

* deluwang23@aliyun.com

**Data Availability Statement:** All relevant data are within the paper and its Supporting Information files.

## Abstract

To investigate the response of blueberry photosynthetic physiology to different light intensities during different stages of fruit development. In this study, four light intensity treatments (25%, 50%, 75% and 100% of full light) were set up to study the change rule of photosynthetic pigment content and photosynthetic characteristics of 'O'Neal' southern highbush blueberry leaves during the white fruiting stage (S1), purple fruiting stage (S2) and blue fruiting stage (S3) under different light intensity environments, and to explore the light demand and light adaptability of blueberry during different developmental stages of the fruit. The results showed that the chlorophyll and carotenoid contents of blueberry leaves showed an increasing trend with decreasing light intensity at all three stages of fruit development. The total chlorophyll content of blueberry leaves at 25% light intensity increased by 76.4% compared with CK during the blue fruiting stage; the maximum net photosynthetic rate (Pmax), light compensation point (LCP), light saturation point (LSP), rate of dark respirations (Rd), inter-cellular $CO_2$ concentration (Ci), stomatal conductance (Gs), transpiration rate (Tr), net photosynthesis rate (Pn), and chlorophyll a/b showed a decreasing trend with decreasing light intensity. The Pn of blueberry leaves was highest under full light conditions at all three stages, and the Pn at 25% light intensity decreased by 68.5% compared with CK during the white fruiting stage Reflecting the fact that blueberries can adapt to low-light environments through increases in chlorophyll and carotenoids, but reduced light intensity significantly inhibited their photosynthesis. The photosynthetic physiology of blueberry showed a consistent pattern at all three stages, but there were some differences in the changes of photosynthetic parameters at different stages. The results of the study can provide theoretical references for the selection of sites and density regulation in blueberry production.

## 1 Introduction

Photosynthesis is the basis of plant growth and development, and is the most important factor constituting productivity, and the photosynthetic performance of plant leaves is positively correlated with production capacity [1]. Light intensity, as one of the main factors affecting plant

**Funding:** This study was funded by the National Natural Science Foundation of China (317602050), and the recipient of the funds was Pro. Delu Wang. The funders had no role in study design, data collection and analysis, decision to publish, or preparation of the manuscript.

**Competing interests:** The authors have declared that no competing interests exist.

photosynthesis, influences the activity of photosynthetic carbon assimilating enzymes, the photoactive opening of stomata, the accumulation of metabolites, and the composition of cyto-chromes. All plants have their own optimal range of light intensity for growth, and too high or too low light intensity will affect plant morphology and photosynthetic physiology [2]. Photo-inhibition often occurs when light intensity is too high, reducing photochemical efficiency and even causing damage to the photo-oxidation system [3]. In addition, low light intensity also affects photosynthesis, severely limiting plant growth and even death [4]. Plants of different species and genotypes have different ranges of adaptation to light intensity and have evolved with a variety of adaptive strategies to minimize the potential damage caused by light stress [5]. Plants can increase light utilisation by lowering the light saturation point and light compensation point to reduce direct absorption of light energy and by lowering stomatal conductance, transpiration rate and intercellular $CO_2$ concentration [8, 9].

Plants can also influence photosynthesis through photosynthetic pigments, which include chlorophyll and carotenoids that are primarily involved in the absorption, transfer, and conversion of light energy. Chlorophyll a is a reaction center pigment that absorbs long-wave light, mainly red light, and emits electrons to two photosystems, P680 and P700, after capturing light energy. Chlorophyll b is a light-trapping pigment that mainly absorbs short-wave light, mainly blue light, and transfers the captured energy to chlorophyll a [6]. Carotenoids help chlorophyll b absorb blue light [7]. Under low light conditions, plants can improve their light utilization capacity and thus better adapt to different light intensities through changes in photosynthetic pigment content [7].

It has now been shown that photosynthetic pigments such as chlorophyll in blueberries increase with decreasing light intensity and photosynthetic parameters such as photosynthetic rate decrease with decreasing light intensity [8–10]. The white, purple and blue fruiting stages are three critical stages for the development of blueberry fruit. However, the response of blueberry photosynthetic physiology to light intensity during different fruit development stage is unknown. Therefore, the study of the effects of light intensity on photosynthetic pigments and photosynthetic characteristics of blueberry during different fruit development stages aims to grasp the light requirements for blueberry fruit growth and development, and to provide a scientific basis for the selection of blueberry stand and density regulation in actual production.

## 2 Materials and methods

### 2.1 Overview of the test site

The experimental site was located in the experimental nursery of the College of Forestry, South Campus of Guizhou University, Huaxi District, Guiyang City, with an elevation of 1159 m, longitude of 104°34′E、latitude of 26°34′N, and a humid and moderate climate in the central subtropics. The annual maximum temperature is 39.5°C, the minimum temperature is -9.5°C, and the annual average temperature is 15.8°C. The annual precipitation is 1229 mm, the average annual relative humidity is 79%, and the total solar radiation is 3567 MJ/m$^2$.

### 2.2 Test materials

The southern highbush blueberry variety 'O'Neill', whose growth potential is basically the same in four years, was used as the test material, and the seedlings for the test were transplanted into plastic pots (diameter of the inner mouth of the pot was 26.5cm, diameter of the bottom was 17.5cm, and the height was 19.7cm), with one seedling in each pot, and the humus soil of the pine forests was used as the substrate for the cultivation.

**Table 1. Actual light intensity corresponding to relative light intensity.**

| Relative light intensity | Actual light intensity($\mu mol\cdot m^{-2}s^{-1}$) | | |
|---|---|---|---|
| | **S1** | **S2** | **S3** |
| 25% | 372±34.06Ad | 369±29.44Ad | 379±28.29Ad |
| 50% | 750±31.18Ac | 699±24.83Ac | 778±30.02Ac |
| 75% | 1123±40.99Ab | 1094±48.50Ab | 1143±36.37Ab |
| CK | 1498±39.26Aa | 1456±44.46Aa | 1587±37.53Aa |

Note: The above table shows the light intensity at 10 a.m. and is measured with a photometer. S1: white fruit stage, S2: purple fruit stage, S3: blue fruit stage. In the table, different uppercase letters indicate significant differences in the same light intensity during different stages, and different lowercase letters indicate significant differences in different light intensity treatments at the same stage ($P < 0.05$), values represent mean ± standard error.

## 2.3 Experimental design

The experiment was set up with four light intensities (relative light intensity, see Table 1), namely 25% (75% shade)、 50% (50% shade)、 75% (25% shade) and 100% light intensity (0% shade), with 100% light intensity as the control (CK), and three groups of replicates were set up for each treatment, with 10 plants per replicate and randomized zonal group placement. Light intensity was controlled using a photometer and a combination of black shade nets with pin numbers of 2, 3, 4, 6, and 8 pins constructed to measure light intensity using a photometer. The experiment was started after blueberry bloom (April 1st).

## 2.4 Sample collection

After one month of shade treatment, blueberry plants with uniform growth and normal fruiting volume were selected, and five to eight plants were used as the source of one biological replicate sample, 28d (white fruit stage S1), 35d (purple fruit stage S2), and 42d (blue fruit stage S3) after full bloom, Three stages of random sampling within the group to pick the tree periphery by the middle of the canopy of uniform size, growth status is basically the same as the blueberry fruit and functional leaves, each time each sample 30g, three stages of a total of three times sampling. The samples were stored directly in liquid nitrogen and brought back to the laboratory to be stored in a -80°C ultra-low temperature refrigerator for use.

## 2.5 Indicator measurement methods

**2.5.1 Determination of photosynthesis-light response curve.** Referring to the method of Gong Zhongzhi [11], the Li-6400 red and blue LED light source was used during the white, purple and blue fruiting stages, and the light intensity was set at 13 levels: 1500, 1300, 1000, 900, 800, 700, 600, 400, 200, 100, 50, 30, and 0$\mu mol m^{-2}s^{-1}$. The photosynthetic physiological parameters such as net photosynthetic rate (Pn), transpiration rate (Tr), stomatal conductance (Gs) and intercellular $CO_2$ concentration (Ci) were measured from the top to the bottom of 4–6 mature functional leaves, and the linear regression analysis was performed between the light intensity (PAR) within 200 $\mu mol m^{-2}s^{-1}$ and the Pn value, and the resulting slope was the apparent quantum efficiency.

*(1) Fitting of PLR.* The PLR (light response curve) was modelled using a single molecular formula [12]:

$$Pn = Pnmax \times [1 - e^{(-AQY \times PAR/Pnmax)}] - Rd \qquad \text{Formula(1)}$$

In the formula: Pn is the instantaneous net photosynthetic rate; Pnmax is the maximum net photosynthetic rate at saturated light intensity; e is a natural constant with a value of about 2.718; AQY is the apparent quantum efficiency; PAR is the photosynthetically active radiation intensity; Rd is the dark respiration rate.

*(2) Calculation of LUE, LCP and LSP.* The LUE (light energy utilization), LCP (light compensation point) and LSP (light saturation point) were calculated based on the PLR curves: LUE is the ratio of Pn to its corresponding PAR under any light intensity; LCP is the PAR value corresponding to when Pn = 0$\mu$molm$^{-2}$s$^{-1}$; LSP is the PAR value corresponding to when Pn reaches 90% of Pnmax; The specific calculation formula is as follows:

$$LUE = Pn \times PAR \qquad\qquad Formula(2)$$

$$LCP = Pnmax \times AQY \times InPnmax \times Pnmax - Rd \qquad\qquad Formula(3)$$

$$LSP = Pnmax \times AQY \times InPnmax \times 0.1Pnmax - Rd \qquad\qquad Formula(4)$$

**2.5.2 Measurement of photosynthetic parameters.**   During the three stages of fruit development, we chose a sunny and windless morning from 8:00 a.m. to 12:00 p.m., and used the Li-6400 red and blue LED light source with a light intensity of 1000 $\mu$molm$^{-2}$s$^{-1}$ to measure the changes of net photosynthetic rate (Pn), intercellular $CO_2$ concentration (Ci), transpiration rate (Tr) and stomatal conductance (Gs) of the leaves during the different developmental stages of the fruits. Three plants were measured for each treatment, and three fixed leaves were measured for each plant, and the stable values were read three times for each measurement, and the mean values were calculated and analysed.

**2.5.3 Determination of chlorophyll content.**   The 80% acetone method was used to determine the chlorophyll content of the leaves of each treatment: 0.1g of leaf blade (cut into pieces) was extracted with 10 mL of 80% acetone in shade until the leaves were completely whitened by the naked eye, and the absorbance values of the treatments were determined in the UV spectrophotometer at the wavelengths of 663 nm, 645 nm, and 470 nm, respectively.

The chlorophyll content was calculated according to the formula (mg/g):

$$Chlorophyll\ a: \left(12.72A_{663} - 2.59A_{645}\right)V/(1\,000W) \qquad\qquad Formula(5)$$

$$Chlorophyll\ b: \left(22.88A_{645} - 4.67A_{663}\right)V/(1\,000W) \qquad\qquad Formula(6)$$

$$Carotenoids: \left(1\,000A_{470} - 3.27Ca - 104Cb\right)V/(229 \times 1\,000\ W) \qquad\qquad Formula(7)$$

$$Total\ chlorophyll: \left(8.05A_{663} + 20.29A_{645}\right)V/(1\,000\ W) \qquad\qquad Formula(8)$$

In the formula: Ca is the concentration of chlorophyll a; Cb is the concentration of chlorophyll b; V is the volume of the extract; W is the fresh weight of the sample.

## 2.6 Data analysis

Excel 2016 and Origin 2024 were used for basic data organization, calculation and graphing; SPSS 26.0 software was used for one-way ANOVA and Duncan's multiple comparisons ($P<0.05$).

# 3 Results and analysis

## 3.1 Effects of different light intensities on photosynthetic pigments in blueberry leaves

As can be seen in Fig 1, light intensity significantly affected the photosynthetic pigment content of blueberry leaves at all three stages. Among the four light intensity treatments, the photosynthetic pigment content of blueberry leaves at all three stages was the lowest in the CK treatment, and its chlorophyll a, chlorophyll b, total chlorophyll, and carotenoids content

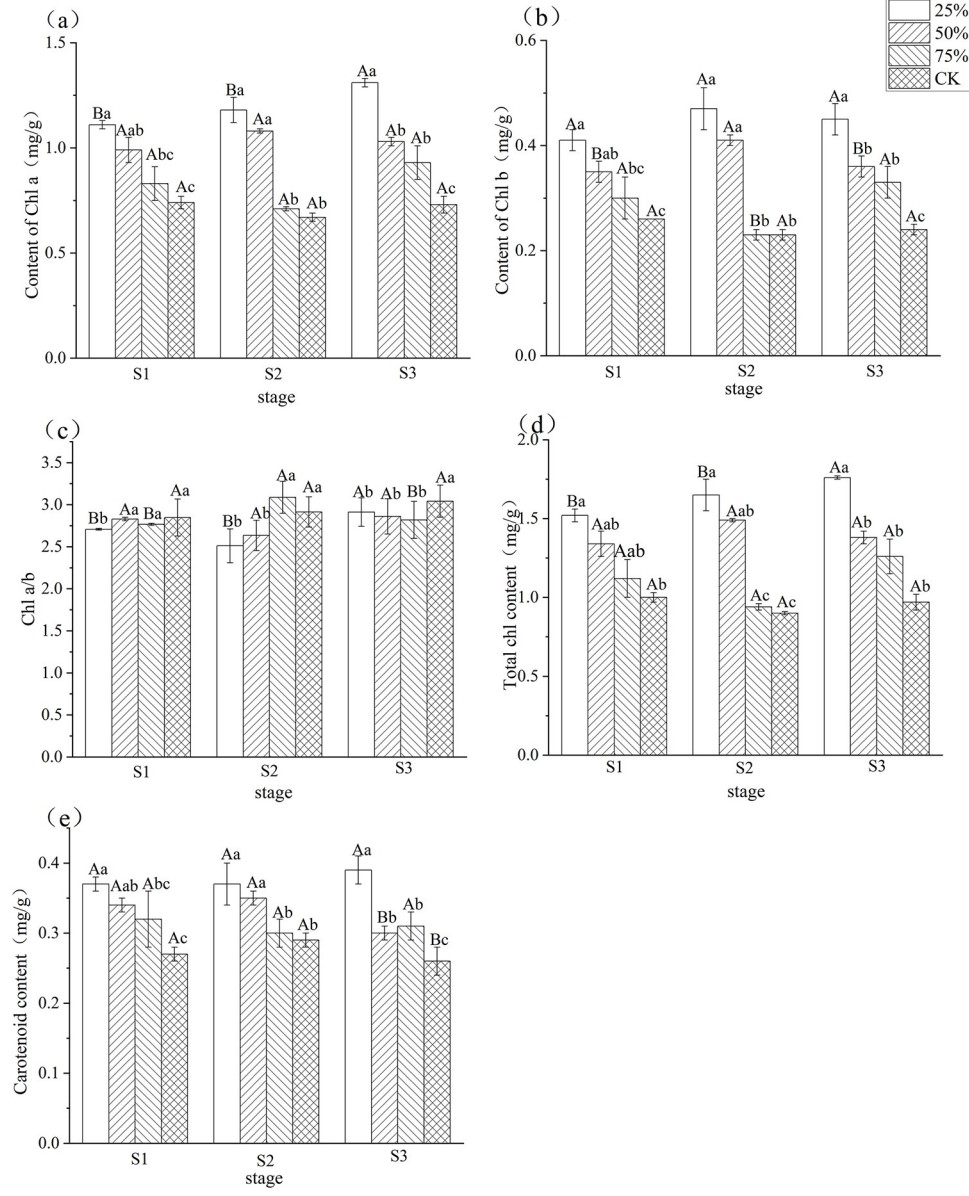

**Fig 1. Effect of different light intensity on photosynthetic pigment of blueberry leaves during different stages.**
Note: In the figure, different uppercase letters indicate significant differences in the same light intensity during different stages, and different lowercase letters indicate significant differences in different light intensity treatments during the same stage ($P< 0.05$). Bars show standard error. S1: white fruit stage, S2: purple fruit stage, S3: blue fruit stage.

increased significantly with decreasing light intensity, and the chlorophyll a/b value decreased. The total chlorophyll, carotenoids, chlorophyll a and chlorophyll b contents from the three stages showed that S3 > S2 > S1. The total chlorophyll, carotenoids, chlorophyll a and chlorophyll b contents of blueberry leaves were the highest at 25% light intensity, which increased by 52.6%, 38.0%, 61.6% and 49.5%, respectively, compared with CK at S1, 65.4%, 37.6%, 82.7% and 59.6% respectively, compared with CK at S2, 76.4%, 44.3%, 77.3% and 76.3% respectively, compared with CK at S3. It can be seen that the chlorophyll and carotenoid contents of low light intensity blueberry leaves were relatively high, indicating that low light promotes the synthesis and accumulation of blueberry chlorophyll and carotenoids, and improves the chlorophyll and carotenoid contents of leaves to improve the light capture capacity and the efficiency of light energy utilisation.

## 3.2 Effect of different light intensities on photosynthetic parameters of blueberry leaves

As shown in Fig 2, in general, intercellular $CO_2$ concentration (Ci), stomatal conductance (Gs), transpiration rate (Tr), and net photosynthesis rate (Pn) of blueberries showed a decreasing trend with decreasing light intensity at the three stages.

Ci reflects the concentration at the instantaneous atmospheric input in dynamic equilibrium with $CO_2$ utilized by plant cell photorespiration and photosynthesis. As shown in Fig 2(A), Ci increased significantly with decreasing light intensity at S1 (except for CK) and decreased significantly at S2 and S3. Gs refers to the degree of stomatal opening. As shown in Fig 2(B), the Gs at all light intensities at all three stages showed that S3 > S2 > S1, and the Gs at 25% light intensity was the smallest, which was 70.1%, 69.6% and 68.1% lower than that of CK, respectively. At the same Stage, Gs decreased with decreasing light intensity. Tr indicates the amount of water lost per unit of leaf area per unit of time and is commonly used to reflect water utilization and metabolism in the plant. As shown in Fig 2(C), Tr of blueberry decreased significantly with decreasing light intensity at all three stages. Tr at 25%, 50% and 75% light intensities were 65.4%, 56.6% and 43.9% lower than CK at S1, 64.1%, 57.0% and 29.0% lower than CK at S2, and 48.9% and 19.3% lower than CK at S3 for 25% and 50% light intensities, respectively. As Gs decreased, plants inhaled less $CO_2$, which also affected Pn. As shown in Fig 2(D), Pn decreased significantly with decreasing light intensity. Compared with CK, Pn of blueberry leaves at 25%, 50% and 75% light intensities was reduced by 68.5%, 53.8% and 33.0% at S1, 61.5%, 53.6% and 30.6% at S2, and 57.0%, 21.4% and 51.6% at S3, respectively. The results showed that light intensity significantly affected blueberry leaves Pn at all three stages of blueberry fruit growth and development, and it gradually decreased with decreasing light intensity.

## 3.3 Effects of different light intensities on photosynthetic-light response parameters in blueberry leaves

As can be seen in Fig 3, the photosynthetic-light response parameters of blueberry leaves were significantly affected by light intensity during the three stages of blueberry fruit growth and development, and generally showed a decreasing trend with decreasing light intensity. Among them, the maximum net photosynthetic rate (Pmax), apparent quantum efficiency (AQY), dark respiration rate (Rd), light compensation point (LCP) and light saturation point (LSP) all showed a significant decreasing trend with decreasing light intensity.

Specifically, there were significant differences in the performance of each photosynthetic-light response parameter index at different light intensities during different stages of fruit development. LSP, LCP, Pmax, and Rd were lowest in blueberry leaves under 25% light intensity treatment, being 44.0%, 24.8%, 64.1%, and 68.3% lower than CK at S1, 40.3%, 17.5%,

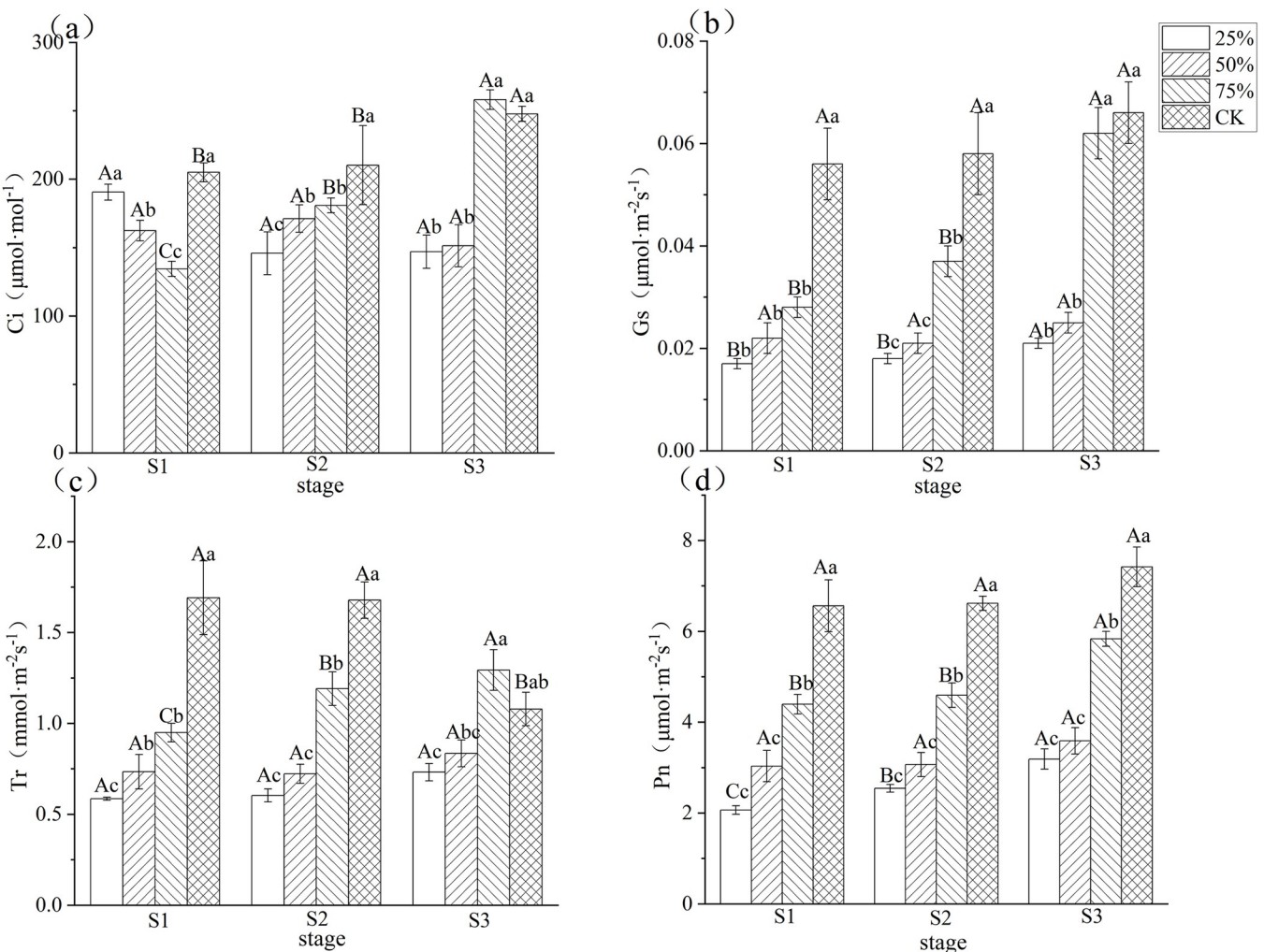

**Fig 2. Effects of different light intensity on photosynthetic parameters of blueberry leaves during different stages.** Note: In the figure, different uppercase letters indicate significant differences in the same light intensity during different stages, and different lowercase letters indicate significant differences in different light intensity treatments during the same stage ($P < 0.05$). Bars show standard error. Ci: intercellular $CO_2$ concentration, Gs: stomatal conductance, Tr: transpiration rate, Pn: net photosynthesis rate. S1: white fruit stage, S2: purple fruit stage, S3: blue fruit stage.

40.8%, and 56.4% lower than CK at S2, 60.3%, 15.6%, 55.4% and 73.3% lower than CK at S3, respectively. It can be seen that blueberries can diminish the LCP, LSP and Rd values under low light intensity environments in order to better utilize the low light for maximally efficient photosynthesis and material accumulation, and at the same time, reduce the consumption of photosynthetically active products to maintain the balance of carbon metabolism in the plant body, so that the plant's growth and development can be carried out normally.

With the increase of light intensity, Pmax, Rd and LSP showed an increasing trend, and the photosynthetic capacity of blueberry leaves was subsequently enhanced to adapt to the stronger light intensity environment. It can be seen that light intensity significantly affected the photosynthetic characteristics of blueberry leaves, and the Pmax of blueberry leaves in CK treatment was significantly increased with the strongest photosynthetic capacity, while Pmax and Rd were significantly decreased in low light intensity, and the plant consumption was reduced, and the photosynthetic capacity was also reduced.

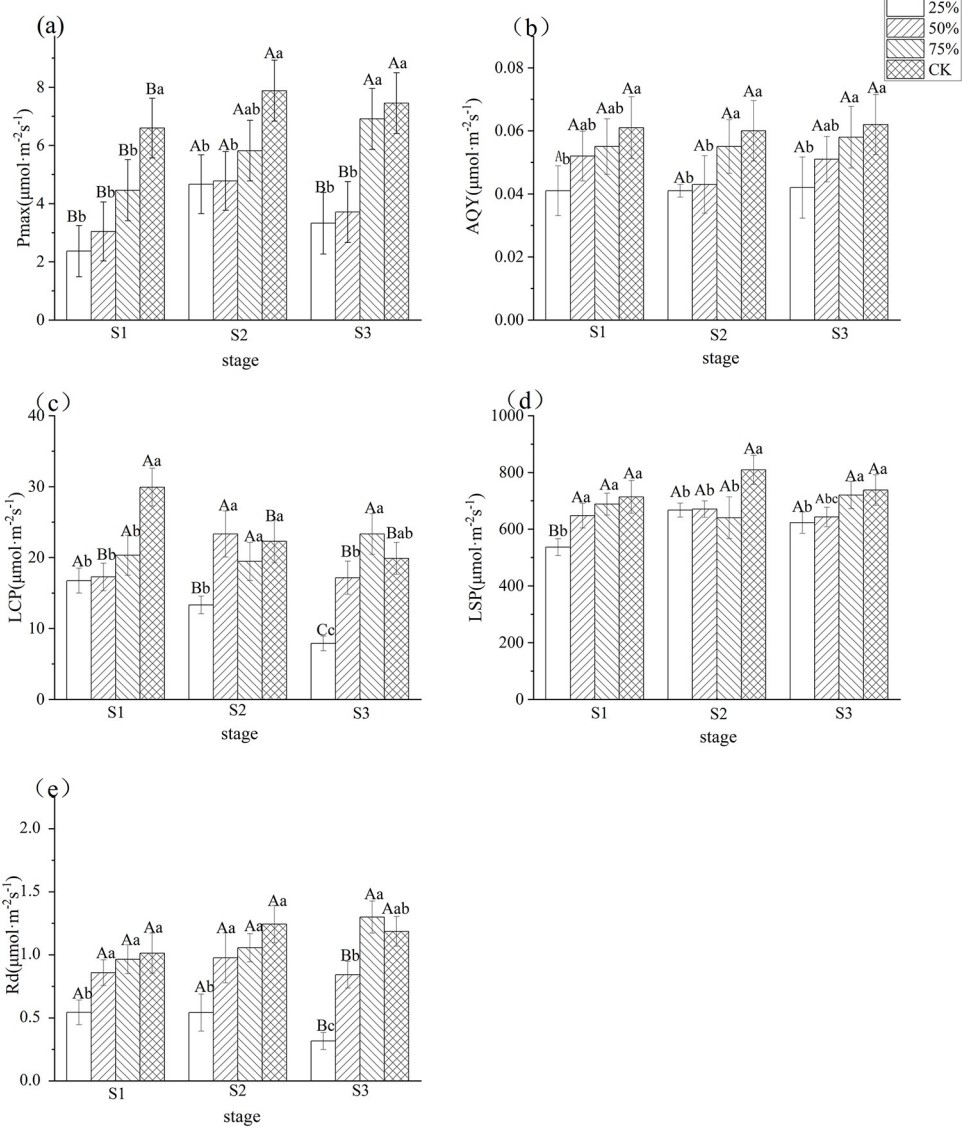

**Fig 3. Effects of different light intensities on photosynthesis-light response parameters of blueberry leaves during different stages.** Note: In the figure, different uppercase letters indicate significant differences in the same light intensity during different stages, and different lowercase letters indicate significant differences in different light intensity treatments during the same stage ($P < 0.05$). Bars show standard error. Pmax: maximum net photosynthetic rate, AQY: apparent quantum efficiency, Rd: dark respiration rate, LCP: light compensation point LSP: light saturation point. S1: white fruit stage, S2: purple fruit stage, S3: blue fruit stage.

## 3.4 Effects of different light intensities on the photosynthetic-light response curve of blueberry leaves

Photosynthetic-light response curves reflect the ability of plants to utilise light intensity. As shown in Fig 4, the net photosynthetic rate (Pn) of blueberry leaves increased with the increase of photosynthetically active radiation intensity (PAR) at all three stages under different light intensity conditions. The increase in Pn gradually slowed down and stabilized after PAR reached 400 μmolm$^{-2}$s$^{-1}$, and as PAR continued to increase, the value of Pn decreased slightly at 1000 μmolm$^{-2}$s$^{-1}$, The three stages showed consistent trends. During the three stages of fruit

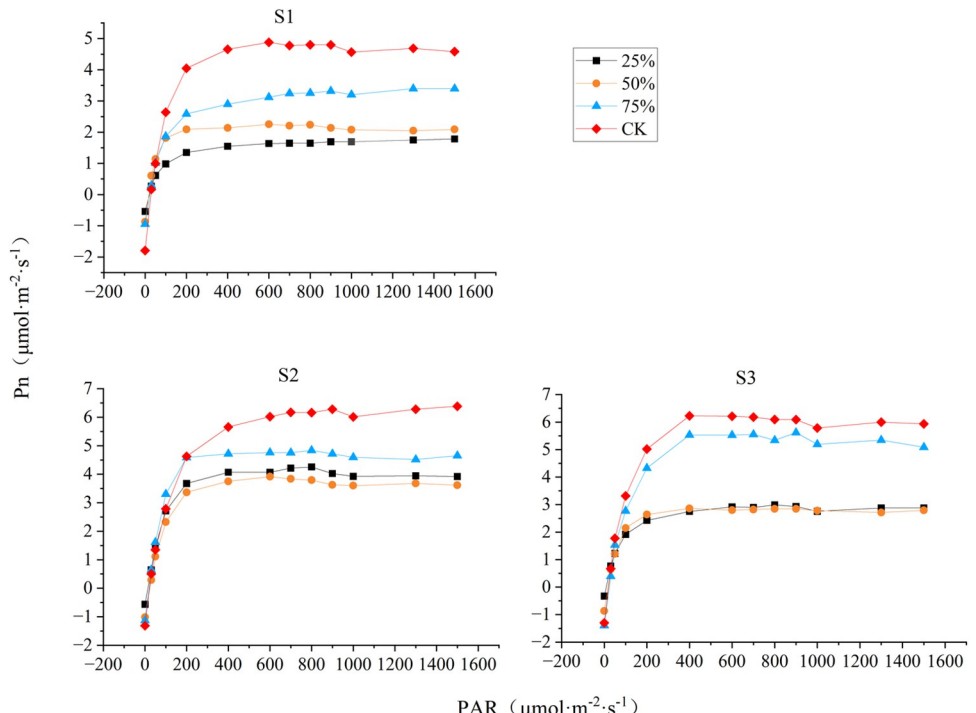

**Fig 4. Effects of different light intensities on the photosynthetic-light response curves of blueberry leaves during different stages.** Pn: net photosynthesis rate, PAR: photosynthetically active radiation intensity. S1: white fruit stage, S2: purple fruit stage, S3: blue fruit stage.

development, CK-treated blueberry leaves had the highest net photosynthetic rate, but the different light intensity treatments did not change the trend of the blueberry photosynthesis-light response curves. It is evident that insufficient light intensity affected the photosynthetic capacity of blueberries.

## 4 Discussion

### 4.1 Decreased light intensity promotes increased photosynthetic pigments in blueberry leaves

Chlorophyll a, chlorophyll b, total chlorophyll and carotenoid contents of blueberry leaves during the three fruit development periods in this study showed an increasing trend with decreasing light intensity. It has been shown that too low a light intensity affects chlorophyll synthesis in plants, and too high a light intensity produces photoinhibition and inhibits chlorophyll synthesis in plants [13]. It has been found that most plants are unable to synthesize chlorophyll in the dark [14]. The chlorophyll content of plant leaves decreases significantly when they are shaded for long periods of time [15]. This is because insufficient light limits the photosynthetic carbon assimilation power and the activity of key enzymes of photosynthesis, resulting in a decrease in the synthesis of chlorophyll in plants.

However, it has also been shown that chlorophyll content increases with decreasing light intensity. In a study on the effect of shade on lingonberry, its chlorophyll content increased with decreasing light intensity [10]. Shade significantly increased the chlorophyll content of summer maize [16], Minnan [17] and dry rice [18]. Studies on sempervirens [19] and gardenia [19] showed that chlorophyll a, chlorophyll b and total chlorophyll of the four species

increased continuously with increasing shade and chlorophyll a/b decreased. Chlorophyll content of the two shrubs also increased with increasing shade [20]. Consistent with the results of the study, the leaves of blueberries showed adaptive changes to low-light stress with significant increases in chlorophyll a, chlorophyll b, carotenoids, and total chlorophyll content during the three growth and development stages, which may be a physiological compensation of the plant for the lack of light intensity. The increase in chlorophyll content under shading conditions may be related to the reduction of photo-oxidative damage, the enlargement of basal lamellae in chloroplasts, and the higher degree of basal lamellae stacking under low light conditions [21]. Chlorophyll a mainly absorbs red light, chlorophyll b mainly absorbs blue-violet light, the proportion of blue light increases and the proportion of red light decreases after the shading treatment, and the proportion changes more significantly with the increase of the degree of shading [22]. The differences in the content of chlorophyll a, chlorophyll b and carotenoids were gradually reduced after shading stress, in which chlorophyll b increased the most, and the value of chlorophyll a/b was reduced to less than 3, which is conducive to the absorption and utilization of blue light in low-light environments, and enhances the adaptive capacity of low-light environments, and the increase in the content and proportion of chlorophyll b is conducive to the maintenance of cystoskeletal membrane integrity [23], and to improve the absorption and utilization of light energy [24].

## 4.2 Diminished light intensity reduces photosynthesis in blueberry leaves

Net photosynthetic rate (Pn), stomatal conductance (Gs), transpiration rate (Tr), and intercellular $CO_2$ concentration (Ci) of blueberry leaves during the three fruit development stages in the study generally showed a decreasing trend with decreasing light intensity. It was shown that reduction in light intensity resulted in lower Pn, Gs, Tr and Ci in soybean plants [25]. Studies on pecan seedlings [26], strawberries [27], alfalfa seedlings [28] and apples [29] yielded consistent findings. A study by Yadong Li et al. [30] on photosynthesis and light intensity in dwarf, semi-highbush and highbush blueberries also showed that their Pn increased with increasing light intensity within this light intensity from the light compensation point to within the light saturation point. Under the conditions of this study, photosynthetic rate and stomatal conductance of blueberry leaves increased with increasing light intensity. This supports previous findings that an increase in stomatal conductance causes rapid changes in photosynthetic rate in response to light conditions [31]. There is a linear relationship between Pn and Gs under a wide range of environmental conditions. Transpiration rate (Tr) and intercellular CO2 concentration (Ci) of blueberry leaves also increased with increasing light intensity in this study. This may be due to the increase in stomatal opening at high light intensity, which increases net $CO_2$ assimilation and water vapor exchange, thereby promoting photosynthesis [32].

The maximum net photosynthetic rate (Pmax), light compensation point (LCP), light saturation point (LSP), apparent quantum efficiency (AQY) and dark respiration rate (Rd) of blueberry leaves during the three fruit development stages in the present study showed a decreasing trend with decreasing light intensity. This is consistent with the results of Zhang Zichuan et al. [8] on lingonberry. The study of photosynthetic properties of four species of Ziziphus spp, by Yannan Wang et al. [33] also showed that LCP, LSP and Rd decreased with decreasing light intensity. It is consistent with the conclusion obtained by Zhang J et al. [34] who studied the changes of LSP and LCP, etc. with light intensity in two maple species. Suggests that relatively low LCP and LSP favor more efficient use of light energy by plants in low light intensity environments, thereby increasing organic matter accumulation. Reduced Rd is often considered an adaptive response for plants to cope with shade conditions and maximize

carbon benefits, suggesting that plants reduce loss of photosynthetics and maintain carbon metabolism homeostasis by lowering Rd as demonstrated, This was also confirmed in the study of *Abies holophylla* [35]. AQY is a measure of the efficiency of light energy conversion in photosynthesis that correctly reflects changes in the functioning of the photosynthetic apparatus and the ability of leaves to utilize low light [36]. The greater the AQY, the more pigment-protein complexes the plant is likely to have for absorbing and converting light energy, and the greater its ability to utilize low light [7]. In this study, AQY gradually decreased with decreasing light intensity, which is similar to the findings of Wang Xiaodong [37], which may be due to the decrease in the number of leaves pigment-protein complexes under low light, resulting in the decrease of AQY in blueberry leaves. Under low light conditions, the photosynthetic capacity of leaves is inhibited and destroyed due to low light stress.

## 5 Conclusion

Four different light intensity treatments had significant effects on the photosynthetic physiology of blueberry. The content of photosynthetic pigments, such as chlorophyll a, increased significantly with decreasing light intensity, and photosynthetic parameters, such as net photosynthetic rate and chlorophyll a/b, decreased significantly. The photosynthetic physiology of the three fruit development stages of blueberry showed similar regularity under different light intensities. Blueberries can adapt to low-light conditions by increasing photosynthetic pigments to improve light energy utilization, but varying degrees of reduced light intensity significantly reduced the photosynthetic efficiency of blueberries. Full light conditions are most favourable to the growth and development of blueberries in all three periods of fruit development. The results show that 'O'Neal' blueberries are light-loving plants and should be grown in a sunny environment and densely planted in practice.

## Supporting information

**S1 Data.**
(XLSX)

## Author Contributions

**Data curation:** Jia Long.

**Formal analysis:** Jia Long, Yunzheng Zhu, Xiaoli An, Xinyu Zhang.

**Funding acquisition:** Delu Wang.

**Investigation:** Tianyu Tan.

**Methodology:** Delu Wang.

**Resources:** Delu Wang.

**Visualization:** Jia Long.

**Writing – original draft:** Jia Long.

**Writing – review & editing:** Delu Wang.

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
