## [Decision Letter · Decision Letter 0]

24 Jul 2024

PONE-D-24-07873Response of blueberry photosynthetic physiology to light intensity during different stages of fruit developmentPLOS ONE

Dear Dr. Wang,

Thank you for submitting your manuscript to PLOS ONE. After careful consideration, we feel that it has merit but does not fully meet PLOS ONE’s publication criteria as it currently stands. Therefore, we invite you to submit a revised version of the manuscript that addresses the points raised during the review process.

We look forward to receiving your revised manuscript.

Kind regards,

Sajid Ali

Academic Editor

PLOS ONE

“National Natural Science Foundation of China.”

3. We note that your Data Availability Statement is currently as follows: [All relevant data are within the manuscript and its Supporting Information files]

Reviewers' comments:

Reviewer's Responses to Questions

**Comments to the Author**

1. Is the manuscript technically sound, and do the data support the conclusions?

Reviewer #1: Partly

Reviewer #2: Yes

2. Has the statistical analysis been performed appropriately and rigorously? 

Reviewer #1: No

Reviewer #2: Yes

3. Have the authors made all data underlying the findings in their manuscript fully available?

Reviewer #1: Yes

Reviewer #2: Yes

4. Is the manuscript presented in an intelligible fashion and written in standard English?

Reviewer #1: No

Reviewer #2: Yes

5. Review Comments to the Author

**Reviewer #1:** The manuscript mainly focuses on the role of light intensity during the different stages of blueberry fruit development. The topic is interesting. However, I have following comments/suggestions/questions to improve the manuscript as listed below:

• What is plastic pruning? Authors haven’t added any information about it. How it will be useful in particular context of this manuscript?

• What is the lesson of this manuscript with respect to density control?

• What are the recommendation for the site selection from the results?

• In table 1, add another top-box and mention S1-S3 stages in it.

• Why authors didn’t consider the fruit quality parameters?

• How many plants were kept under each covering?

• Line 93 need correction. “The annual cumulative temperature above 10 ℃ is 4637.5 ℃”

• What was the name of cultivar?

• Statistical analysis need to be revise. In figure 1-a and -b, the statistical groups for S1 (lowercase) are not satisfactory.

• In figure 3, all abbreviations need to be explained in the caption.

• Graph quality can be improved by enhancing the font size.

**Reviewer #2: **In the manuscript PONE-D-24-07873, the authors investigated response of blueberry photosynthetic physiology to light intensity during different stages of fruit development. It is my opinion that there is some useful information in the manuscript. However, the following points may be considered for improving the quality of this manuscript.

Abstract

The sentence should be improved "To investigate the response of blueberry photosynthetic physiology to different light intensities during different stages of fruit development." Some details, such as the specific percentages of chlorophyll content increase, could be summarized more concisely.

Introduction

Some information is repeated, such as the effects of light intensity on photosynthesis. Some statements, such as "light is the source of energy for photosynthesis in plants," are overly basic and could be omitted or elaborated with specific relevance to blueberries.

How does the light intensity range used in the study compare to the known optimal range for blueberries?

Are there any previous studies specifically on blueberry photosynthetic response to light intensity that could provide a more focused context?

Materials and Methods

The formulas for calculating photosynthetic parameters are presented without sufficient context or explanation for readers who might not be familiar with them. The description of the light intensity treatments could be more concise. How was the relative light intensity verified throughout the experiment?

Were there any steps taken to ensure that this storage method did not affect the chlorophyll or photosynthetic measurements?

Discussion

The discussion mentions that low light intensity affects chlorophyll synthesis negatively but then states that chlorophyll content increases with decreasing light intensity. This contradiction needs to be addressed and clarified. The explanation of stomatal and non-stomatal limitations on photosynthesis is somewhat confusing. It would benefit from a clearer distinction between the two and how each was assessed in this study. The discussion could delve deeper into the potential mechanisms behind the observed changes in photosynthetic parameters and pigment content under different light intensities.

Conclusion

The conclusion could provide more specific recommendations based on the study's findings, such as optimal light conditions for different stages of fruit development.

**Editor Comments: **The comments and concerns of the Reviewer 1 are very genuine and critical. Please address these comments. The revision will not be the guarantee for acceptance of the article.** **

6. PLOS authors have the option to publish the peer review history of their article (what does this mean?). If published, this will include your full peer review and any attached files.

Reviewer #1: No

Reviewer #2: **Yes: **Ghulam Khaliq

---

## [Author Response · Author response to Decision Letter 0]

16 Aug 2024

Responses to reviewer 1:

1. the information on shaping and pruning has been removed.

2. regarding the suggestion on density control, the conclusion has been revised and pointed out that plants should be reasonably spaced to avoid shading each other.

3. Site selection is recommended to choose sunny and shade-free areas. (Modified conclusions have been noted)

4. The recommendation on Table 1 has been modified.

5. Concerning the fruit quality parameters lack of consideration in this study, our group will further improve it in the future.

6. The article has pointed out 3 sets of replications for each treatment, where each replication is 10 plants, i.e.: 30 blueberry plants for each treatment.

7. The section (line 93) has been amended.

8. The article has indicated that the variety is called 'O'Neill'.

9. the statistical analysis has been revised.

10. All abbreviations in the revised graph have been explained in the title.

11. fonts have been adjusted to improve graphic quality.

 Thank you for your valuable suggestions!

Responses to reviewer 2:

1 Abstract

 Has been improved to a more concise summary.

2 Introduction

 Repetitive information about the introduction has been fixed. The setup of the light intensity range in this study covers a wide range of light intensities and can be illustrative. Few studies have been conducted specifically on the photosynthetic response of blueberries to light intensity, so the context is not as focused.

3 Materials and Methods

 The formulas of the revised article on photosynthetic parameters have provided the background. Relative light intensity was controlled with shade nets of different densities. The method of ultra-low temperature storage had less effect on chlorophyll content. Measurements of photosynthetic parameters were determined with a photosynthesizer at the experimental base.

4 Discussion

 The discussion mentions that in previous studies: low light intensity has both negative and positive effects on chlorophyll synthesis, while the article points out that in this study low light intensity had a positive effect on chlorophyll synthesis. The question about stomatal conductance has been revised. The reason behind the changes in photosynthetic parameters and photosynthetic pigments under different light intensities has been added.

5 Conclusion

 Based on the suggested modifications, the conclusion has indicated the optimum light conditions for different stages of fruit development.

 Thank you for your valuable suggestions on this article!

---

## [Decision Letter · Decision Letter 1]

28 Aug 2024

Response of blueberry photosynthetic physiology to light intensity during different stages of fruit development

PONE-D-24-07873R1

Dear Dr. Wang,

We’re pleased to inform you that your manuscript has been judged scientifically suitable for publication and will be formally accepted for publication once it meets all outstanding technical requirements.

Kind regards,

**Sajid Ali**

Academic Editor

PLOS ONE

Additional Editor Comments (optional):

Reviewers' comments:

Reviewer's Responses to Questions

**Comments to the Author**

1. If the authors have adequately addressed your comments raised in a previous round of review and you feel that this manuscript is now acceptable for publication, you may indicate that here to bypass the “Comments to the Author” section, enter your conflict of interest statement in the “Confidential to Editor” section, and submit your "Accept" recommendation.

Reviewer #1: All comments have been addressed

Reviewer #2: All comments have been addressed

2. Is the manuscript technically sound, and do the data support the conclusions?

Reviewer #1: Yes

Reviewer #2: Yes

3. Has the statistical analysis been performed appropriately and rigorously? 

Reviewer #1: Yes

Reviewer #2: Yes

4. Have the authors made all data underlying the findings in their manuscript fully available?

Reviewer #1: Yes

Reviewer #2: Yes

5. Is the manuscript presented in an intelligible fashion and written in standard English?

Reviewer #1: Yes

Reviewer #2: Yes

6. Review Comments to the Author

Reviewer #1: (No Response)

Reviewer #2: After revision, the authors have adequately addressed most of the issues raised during the review process. I am pleased to see that the revised manuscript has significantly improved. With these revisions, I now consider the manuscript to be acceptable for publication. I recommend accepting the manuscript for publication in its current form.

7. PLOS authors have the option to publish the peer review history of their article (what does this mean?). If published, this will include your full peer review and any attached files.

Reviewer #1: **Yes: **Rana Naveed Ur Rehman

Reviewer #2: **Yes: **Ghulam Khaliq

---

## [Editor Report · Acceptance letter]

15 Sep 2024

PONE-D-24-07873R1 

PLOS ONE

Dear Dr. Wang, 

I'm pleased to inform you that your manuscript has been deemed suitable for publication in PLOS ONE. Congratulations! Your manuscript is now being handed over to our production team.

Kind regards, 

on behalf of

Dr. Sajid Ali 

Academic Editor

PLOS ONE